# Molecular Mechanisms of Craniofacial and Dental Abnormalities in Osteopetrosis

**DOI:** 10.3390/ijms241210412

**Published:** 2023-06-20

**Authors:** Yu Ma, Yali Xu, Yanli Zhang, Xiaohong Duan

**Affiliations:** 1College of Life Sciences, Northwest University, Xi’an 710069, China; 18391689867@163.com (Y.M.); a2258879785@126.com (Y.X.); 2State Key Laboratory of Military Stomatology, National Clinical Research Center for Oral Disease, Shaanxi Key Laboratory of Stomatology, Department of Oral Biology & Clinic of Oral Rare Diseases and Genetic Diseases, School of Stomatology, The Fourth Military Medical University, Xi’an 710032, China

**Keywords:** osteopetrosis, osteoclast, tooth, craniofacial bone

## Abstract

Osteopetrosis is a group of genetic bone disorders characterized by increased bone density and defective bone resorption. Osteopetrosis presents a series of clinical manifestations, including craniofacial deformities and dental problems. However, few previous reports have focused on the features of craniofacial and dental problems in osteopetrosis. In this review, we go through the clinical features, types, and related pathogenic genes of osteopetrosis. Then we summarize and describe the characteristics of craniofacial and dental abnormalities in osteopetrosis that have been published in PubMed from 1965 to the present. We found that all 13 types of osteopetrosis have craniomaxillofacial and dental phenotypes. The main pathogenic genes, such as chloride channel 7 gene (*CLCN7*), T cell immune regulator 1 (*TCIRG1*), osteopetrosis-associated transmembrane protein 1 (*OSTM1*), pleckstrin homology domain-containing protein family member 1 (*PLEKHM1*), and carbonic anhydrase II (*CA2*), and their molecular mechanisms involved in craniofacial and dental phenotypes, are discussed. We conclude that the telltale craniofacial and dental abnormalities are important for dentists and other clinicians in the diagnosis of osteopetrosis and other genetic bone diseases.

## 1. Introduction

Osteopetrosis refers to a group of rare hereditary skeletal diseases, characterized by increased bone density and frequent fractures, which is also known as “Albers-Schönberg disease”, following the name of Dr. Albers Schönberg, a German radiologist who first described this bone disease in 1904. According to its inheritance mode, osteopetrosis can be divided into three categories: autosomal recessive osteopetrosis (ARO), autosomal dominant osteopetrosis (ADO), and X-linked osteopetrosis (XLR) [1]. The Online Mendelian Inheritance in Man (OMIM) website subgroups osteopetrosis into 13 main types based on clinical appearance, genetic pattern, and causative genes. ARO and ADO can be further divided into nine types (Osteopetrosis, Autosomal Recessive 1~9) and three types (Osteopetrosis, Autosomal Dominant 1~3), respectively. Patients with XLR osteopetrosis often have symptoms of ashidrotic ectodermal dysplasia with immune deficiency and lymphedema, which Doffinger et al. [2] named ashidrotic ectodermal dysplasia with immune deficiency, osteopetrosis, and lymphedema (OLEDAID) in 2001. 

According to the published data and our analysis, at least 11 genes are correlated with osteopetrosis. These genes may be divided into five broad groups. The first group mainly contains tumor necrosis factor ligand superfamily, member 11 (*TNFSF11*, RANKL) and its receptor tumor necrosis factor receptor superfamily 11A (*TNFRSF11A*, RANK), which regulate osteoclastogenesis. The second group contains the genes encoding ion channels, transporters, or pH sensors, such as *TCIRG1*, *CLCN7*, *OSTM1*, sorting nexin 10 (*SNX10*), solute carrier family 4 member 2 (*SLC4A2*), and *PLEKHM1*, which may control osteoclastic acidification. The third group of genes, including *LRP5*, may increase Wnt canonical signaling to enhance osteoblast activity. The fourth group belongs to the cytoplasmic enzyme group, which is encoded by *CA2* and catalyzes the formation of HCO3 ^−^ and H^+^ ions. Nuclear factor kappa-b kinase, regulatory subunit gamma (*IKBKG*) is an NF-κB essential modulator.

Our previous findings demonstrated that the first symptom of craniofacial bone and tooth dysplasia is quite common in osteopetrosis patients [3]. Owing to the highly overlapping phenotypes seen in individuals classified with osteoclasts, there is some confusion as to how persons with the skeletal phenotypes of osteopetrosis should be classified. 

In this review, osteopetrosis-related references were collected from PubMed (1965 to the present), with keywords including osteopetrosis, osteomyelitis, mandible, maxilla, tooth, craniofacial, skull, and calvarium. The clinical features, types, and related pathogenic genes of osteopetrosis are summarized and discussed, especially the craniofacial bone and tooth phenotypes, and their underlying pathogenic genes.

## 2. Genotype and Clinical Phenotype of Osteopetrosis

### 2.1. Osteopetrosis, Autosomal Dominant 1

Osteopetrosis, Autosomal Dominant 1 (OPTA1, OMIM #607634), is relatively mild, characterized by generalized osteosclerosis, especially in the cranial roof. Other clinical features are diffuse sclerosis found in the pelvis without endobones, no thickening of the spinal endplate, symmetrical osteosclerosis and progressive presentation with age, and diffuse osteosclerosis of the cortical and trabecular bones. OPTA1 is the only type of osteopetrosis without frequent fractures [4]. Craniofacial features include significant sclerosis and thickening of the cranial vault, and frontal sinuses obliteration. A 56-year-old female patient, reported on by Pangrazio [5], showed increased bone density in the mandible, skull base, and legs. She also experienced sudden blindness in the left eye, unilateral hearing impairment, visual impairment in the right eye, and worsening hearing problems; in addition, she complained of frequent headaches. OPTA1 is caused by a heterozygous mutation in the low-density lipoprotein receptor-related protein 5 gene (*LRP5*) [4].

### 2.2. Osteopetrosis, Autosomal Dominant 2

Osteopetrosis, Autosomal Dominant 2 (OPTA2, OMIM #166600), was the first type of osteopetrosis recognized, and is also known as Albers-Schönberg disease. OPTA2 generally develops in late childhood or early adulthood and exhibits a milder phenotype, and is therefore classified as a “benign” form of osteopetrosis, with a frequency of about 5.5 in 100,000 people [6]. The clinical features mainly include a high rate of fracture, segmentary osteosclerosis, osteomyelitis, endobones, and scoliosis, as well as vision loss and bone marrow failure [7]. Autosomal dominant osteopetrosis is divided into two distinct types [8]. Type I shows an obvious sclerosis of the skull with an increased thickness of the cranial wall. Sclerosis of the skull in type II is most pronounced at the base. Moreover, type II shows a typical ‘rugger-jersey spine’ and endobones are seen in the pelvis. In craniofacial clinical manifestations, OPTA2 has a relatively common base sclerosis of the skull, mandibular osteomyelitis, dental deformity, and cranial nerve compression [3]. Seventy percent of the cases of autosomal dominant osteopetrosis are OPTA2, which is the most common form of osteopetrosis, and caused by a heterozygous mutation in the chloride channel 7 gene (*CLCN7*) [9]. OPTB4 is also caused by a *CLCN7* mutation. Most *Clcn7*-deficient mice exhibit severe phenotypes, and it can even be life-threatening for a homozygous mouse [10,11,12,13]. High bone density, neurodegeneration, dental abnormalities, and retinal degeneration are similar to the human phenotype blemish [10,11,12,13,14,15,16,17,18,19].

### 2.3. Osteopetrosis, Autosomal Dominant 3

Osteopetrosis, Autosomal Dominant 3 (OPTA3, OMIM #618107), is characterized by variable phenotypes. Some patients have relatively malignant clinical phenotypes of osteopetrosis, including fractures after minor trauma, missing teeth, significant anemia and hepatosplenomegaly, and a generalized increased bone density throughout the body [20], while others exhibit osteosclerosis of the skull and generalized osteopenia [21]. OPTA3 is caused by a heterozygous mutation in the pleckstrin homolog domain, containing the family M (with run domain) member 1 (*PLEKHM1*) gene [21]. A *Plekhm1* conditional knockout mouse shows an increase in trabecular bone mass, but no obvious abnormalities in its other organs [22]. Thus, the lack of an animal model hampers research into OPTA3.

### 2.4. Osteopetrosis, Autosomal Recessive 1

Osteopetrosis, Autosomal Recessive 1 (OPTB1, OMIM #259700), belongs to infantile malignant osteopetrosis, and the survival age of patients is generally less than ten years old. The clinical characteristics of patients are generally macrocephaly, forehead protrusion, blindness, hepatosplenomegaly, anemia, motor retardation, and skeletal sclerosis. Frattini et al. [23] reported on five patients. Patients 1, 3, and 4 showed visual impairment and hepatosplenomegaly, and osteoclasts were found in their bone marrow. Patients 2 and 5 developed anemia and thrombocytopenia. Four of them had bone marrow transplants (BMT), but only two of the five patients survived. Typical craniofacial phenotypes include macrocephaly, anterior fontanel prominence, forehead eminence, and skull base and craniofacial bone thickening [24]. The tooth phenotypes mainly include dentition defect, tooth loss, tooth development delay, tooth deformity, and alveolar bone abnormality [25]. OPTB1 is caused by a heterozygous or homozygous mutation in the gene encoding *TCIRG1* [23]. A *Tcirg1-*deficient mouse shows the typical osteopetrosis phenotype of humans, including abnormal tooth eruption and morphology, skull abnormalities, increased bone mass, and a shortened life [26,27,28,29,30,31].

### 2.5. Osteopetrosis, Autosomal Recessive 2

Osteopetrosis, Autosomal Recessive 2 (OPTB2, OMIM #259710), a mild autosomal recessive osteopetrosis, was first reported in 1978 [32], and later became known as autosomal recessive osteopetrosis (ARO) type II, or OPTB2. Its common clinical features include hepatosplenomegaly, increased bone density throughout the body, thickened bone cortices, such as of the spine, leg bones, and pelvis, and being prone to fractures. In addition, Sharma et al. [33] reported on a patient with hydrocephalus in the brain, blindness in the left eye, facial bone sclerosis, and facial deformity. OPTB2 has typical craniofacial deformities, including a forehead bulge, increased head circumference, skull base hardening or thickening, mandibular prognathism, facial deformity, and osteomyelitis of the jaw. At the same time, dental deformities are the typical feature of OPTB2, including dental caries, root or crown deformities, retained deciduous teeth, delayed eruption of the permanent teeth, and alveolar bone defects. OPTB2 is caused by a mutation of the *TNFSF11* gene [34]. A *Tnfsf11*-deficient mouse exhibits the typical osteopetrosis changes, which may provide the experimental data and theoretical foundation for the study of craniofacial and dental phenotypes [35,36,37,38,39]. 

### 2.6. Osteopetrosis, Autosomal Recessive 3

Osteopetrosis, Autosomal Recessive 3 (OPTB3, OMIM #259730), belongs to infantile malignant autosomal recessive osteopetrosis. Its main characteristic is osteopetrosis with renal tubular acidosis. The main clinical manifestations are short stature, facial feature abnormalities, chest and finger deformities, diffuse osteopetrosis, fracture, mental retardation with inattention, and intellectual impairment. Pang et al. [40] reported on two cases with physical and mental retardation and increased bone density. One of them had thoracic deformities, including a barrel chest, rachitic rosary, and costal margin valgus. An X-ray examination showed a generalized increased bone density, a thickened cortical bone, and a ‘Sandwich’ appearance of the vertebrae. Craniofacial phenotypes are characterized by abnormal facial features, including small mandibles, malocclusion, a high-arch palate, dental crowding, disordered teeth, and dental caries. In one report, the patient showed mental retardation and limb weakness, and dental examination results showed more caries and tooth dislocation [41]. Fathallah et al. [42] traced the origins of 24 Tunisian families with *CAII* deficiency, and found they were descended from a common ancestor from the Arabian Peninsula. OPTB3 is caused by a homozygous or compound heterozygous mutation in the gene encoding carbonic anhydrase II (*CA2*) [42]. *Ca2*-deficient mice have relatively complex phenotypes, and only some of them present with increased bone mass and tubular acidosis [43,44].

### 2.7. Osteopetrosis, Autosomal Recessive 4

Osteopetrosis, Autosomal Recessive 4 (OPTB4, OMIM #611490), belongs to infantile malignant autosomal recessive osteopetrosis. So far, there have been three reported cases of this type of osteopetrosis [6,45]. Its severe clinical phenotypes include fractures with minor trauma and severe osteopetrosis, reticulocytosis, anemia, optic nerve atrophy, hepatosplenomegaly, abnormal medullary bone formation, and narrowing of the medullary. The craniofacial phenotypes include sclerosis, particularly of the base of the skull. All three children underwent BMT, but all died by the age of two. OPTB4 is caused by a homozygous or compound heterozygous mutation in the *CLCN7* gene [6].

### 2.8. Osteopetrosis, Autosomal Recessive 5

Osteopetrosis, Autosomal Recessive 5 (OPTB5, OMIM #259720), belongs to infantile malignant osteopetrosis. It has the most severe clinical phenotype, with some cases experiencing symptoms during the fetal period, and most die within the first few years, even within a year. Jean Vacher [46] showed its main characteristic is osteopetrosis with neurodegeneration. The most frequent manifestations are skull foramina leading to developmental delay, hydrocephalus, cerebral atrophy, visual dysfunction, and spasticity. This type of patient has a typical craniofacial phenotype, including a ‘harlequin mask’ or ‘space alien face’ appearance, prominent eyebrows, a sclerotic skull base, thickening of the calvaria, hypoplastic mandible, and abnormal dental development [47,48,49]. OPTB5 is caused by a homozygous mutation in the gene encoding osteopetrosis-associated transmembrane protein1 (*OSTM1*) [46]. An *Ostm1*-deficient mouse exhibits high bone density, abnormal bone morphology, dental abnormalities, etc. The most common brain lesions in OPTB5 are also present in the mouse model, and abnormalities in the cerebellum may be caused by skull damage [50,51,52].

### 2.9. Osteopetrosis, Autosomal Recessive 6

Osteopetrosis, Autosomal Recessive 6 (OPTB6, OMIM #611497), is an intermediate type. Van et al. [53] reported on a girl and her youngest brother, and identified a homozygous G > A transition at position +1 of a splice site in intron 3 of the *PLEKHM1* gene. Her unaffected parents and a clinically and radiographically normal brother were heterozygous for the mutation. At the age of seven, she had an ‘Erlenmeyer flask’ deformity of the distal femora. At the age of 14, the patient had pain in her left leg when walking and was diagnosed with chondrolysis of the left hip. An X-ray showed cortical sclerosis of the pelvis, particularly the iliac wings. The patient’s vertebral endplates showed band-like sclerosis and distal femora, and the proximal fibulae and tibiae showed uneven sclerosis. As for the craniofacial phenotypes, no relevant cases have been reported. According to the published results, OPTB6 is caused by a mutation of the *PLEKHM1* gene, the same as OPTA3 [53].

### 2.10. Osteopetrosis, Autosomal Recessive 7

Osteopetrosis, Autosomal Recessive 7 (OPTB7, OMIM #612301), belongs to malignant infantile osteopetrosis. In this kind of osteopetrosis, with a clinical diagnosis based on radiologic and hematologic defects, it is identified through genetic detection. It is usually detected during infancy or early childhood due to whole-body bone sclerosis, hepatosplenomegaly, severe anemia, persistent fracture, increased head circumference, and tooth deformities. Xu et al. [54] reported on a 10-year-old girl with generalized sclerosis, including thickening and enlargement of the skull plate and a typical ‘Sandwich’ vertebra. Guerrini et al. [55] reported on eight patients from seven families, four of whom had reduced serum immunoglobulin levels. The most common phenotype is nystagmus or visual impairment, some with skull plate thickening or skull deformity, and delayed tooth eruption or root malformation. Taylor-Miller et al. [56] reported on a male patient with markedly delayed primary dentition on examination, mild facial asymmetry, and whose ophthalmological assessment revealed nystagmus, optic nerve pallor, and left visual impairment. OPTB7 is caused by a homozygous or compound heterozygous mutation in the tumor necrosis factor receptor superfamily 11A (*TNFRSF11A*) gene [56]. Affected animals have a severe osteopetrosis phenotype, including shortened long bones, metaphyseal widening, doming of the skull, failure to erupt teeth, and increased bone density [57,58,59,60].

### 2.11. Osteopetrosis, Autosomal Recessive 8

Osteopetrosis, Autosomal Recessive 8 (OPTB8, OMIM #615085), belongs to infantile malignant autosomal recessive osteopetrosis. The clinical features include anemia, thrombocytopenia, hepatosplenomegaly, increased bone density at the epiphyses and metaphyses, and increased thickness of the vertebrae, showing a ‘Sandwich’ shape of the lamina. Udupa et al. [61] reported on a 4-year-old patient with a large skull, pectus carinatum, and frontal bossing. His diaphyseal bone density was irregular, his vertebral body was thickened, his lamina was ‘Sandwich’ shape, and his blood tests showed anemia and thrombocytopenia. He had vision problems, hepatosplenomegaly, and mesenteric lymphadenopathy at five years of age. The craniofacial phenotype includes increased head circumference, frontal bossing, a broad open fontanelle, severe optic atrophy, vision impairment and even blindness, and hearing impairment. Mégarbané et al. [62] showed that a proband had macrocephaly, anemia, and splenomegalia at four months of age. At two years of age, the proband displayed an open anterior fornix, a triangular face, fronto-occipital protrusion, eyeball protrusion, strabismus, and a small chin. His older brother lost his sight at the age of three and died of hydrocephalus at the age of five. His cousin, with similar clinical features, is blind and deaf. OPTB8 is caused by a homozygous mutation in the sorting nexin 10 (*SNX10*) gene [61]. A *Snx10*-deficient mouse exhibits typical osteopetrosis features, including high bone density and a “bone-in-bone” appearance, as well as failed tooth eruption and abnormal jaw bones [63,64,65].

### 2.12. Osteopetrosis, Autosomal Recessive 9

Osteopetrosis, autosomal recessive type 9 (OPTB9, OMIM #620366), is typically characterized by increased bone density, cortical sclerosis of the proximal femur, and increased bone fragility. Craniofacial features are characterized by significant sclerosis of the cranial bones and skull base, and visual impairment due to compression of the narrow optic nerve. Xue et al. [66] reported on a 58-year-old female patient who presented with skeletal pain and increased bone density, with spontaneous fractures of the tibia and fibula, and sclerotic bones of the cranial skull, skull base, and trunk. The patient had poor vision due to bilateral papilledema, due to compression of the optic nerve canal. She also had progressive renal failure and hyperparathyroidism. OPTB9 is caused by a complex heterozygous mutation of the solute carrier family 4 member 2 (*SLC4A2*) gene [66,67]. The clinical descriptions of this patient showed craniofacial defects, which were similar to those observed in *Slc4a2*-deficient red angus cattle and mice. The OPTB9 patients did not show obvious dental phenotypes; however, *Slc4a2*-deficient animal models result in dental problems [66].

### 2.13. Anhidrotic Ectodermal Dysplasia Associated with Immune Deficiency, Osteopetrosis, and Lymphedema

Anhidrotic ectodermal dysplasia associated with immune deficiency, osteopetrosis, and lymphedema (OLEDAID, OMIM #300291), was named by Doffinger et al. in 2001 [2]; it has an X-linked recessive genetic pattern. The typical features include conical incisors, frontal bossing, hypo/anhidrosis, thin skin or hair, lymphedema, and osteopetrosis. Patients generally die of immune deficiency or bacterial infection. OLEDAID is caused by a semi-zygous mutation or deletion of the *IKBKG* gene on Xq28, with a genotype–phenotype correlation in hemizygous males. Moreover, loss-of-function mutations and hypomorphic mutations cause incontinentia pigmenti in females [68,69]. Swarnkar et al. [70] reported on two myeloid-specific *Ikbkg*-deletion mouse models (NM-cKO–LysM and NM-cKO–CTSK), both of which exhibited an osteopetrotic phenotype, and accompanied by splenomegaly and growth retardation. 

## 3. Molecular Pattern of Osteopetrosis

### 3.1. RANKL and RANK

Tumor Necrosis Factor Ligand Superfamily, Member 11 (*TNFSF11*, RANKL) and Tumor Necrosis Factor Receptor Superfamily, Member 11A (*TNFRSF11A*, RANK) mutations cause OPTB2 and OPTB7, respectively. RANKL is mainly produced by osteoblasts and binds to the membrane of osteoblasts, whereas RANK is mainly expressed by hematopoietic cells and osteoclasts and their precursors, and is a specific receptor of RANKL. RANKL stimulates monocytes to differentiate into osteoclasts by activating its receptor, RANK. Activation of RANK can promote the differentiation and maturation of osteoclasts, increasing the survival time of osteoclasts, and activating bone resorption capacity through the NF-κB pathway, PI3k-AKT pathway, MAPK pathway, and other pathways. In addition, the binding of RANKL and RANK can be competitively blocked by OPG to regulate the production and activity of osteoclasts, thus regulating the construction and remodeling of bone. Previous research has suggested that, unlike RANKL-deficient OPTB2, which cannot be cured by hematopoietic stem cell transplantation (HSCT), RANK-deficient OPTB7 can be cured by HSCT [1].

*Tnfrsf11a^75dup27/75dup27^* homozygotes exhibited osteopetrosis and their bone marrow cells were unable to form osteoclasts under the stimulation of RANKL and M-CSF [57], which was caused by reducing IκB and p38 activation. Qiu et al. [71] revealed a novel insight into human M199-induced ARO. Changes in the binding of M200s to their receptor RANK or sabotaged trimerization affected the marker gene and downstream signaling cascades, including NF-κB, NFATc1, and ROS. In *Rankl* knockout mice, both osteoclasts and osteoblasts were deficient and it was difficult to observe the osteoblasts even after successful differentiation [72]. Huang et al. [38] found that osteoclasts are important for odontoblast differentiation and tooth root formation and eruption, possibly through IGF/AKT/mTOR signaling, in *rankl*-deficient mice. 

### 3.2. TCIRG1

T Cell Immune Regulator 1 (*TCIRG1*) belongs to the vacuolar H^+^-ATPase (V-ATPase) family, encoding the a3 subtype of V-ATPase, which belongs to the proton translocation domain V0, and is mainly responsible for proton transport in osteoclasts. TCIRG1 is crucial for the fusion of osteoclast precursors and the function of mature osteoclasts. More than 50% of the cases of human malignant infant osteopetrosis are caused by *TCIRG1* mutations [1]. *TCIRG1* mutations commonly cause bone mineralization defects and the co-occurrence of osteopetrosis and rickets, because V-ATPase keeps a low pH for bone resorption in bones and for Ca^2+^ absorption in the stomach [73]. Patients with *TCIRG1* mutations usually have the same or an increased number of osteoclasts, but a loss of osteoclast function. In 2020, Palagano et al. [26] created a mouse model with *Tcirg1^oc^*-deficient autosomal recessive osteopetrosis (NSG oc/oc). In NSG oc/oc mice, osteoclasts formed, but most of them were small and irregularly-shaped, and bone resorption was impaired. Zhang et al. [74], in a study of the knockdown of *Tcirg1* in mouse bone marrow-derived monocytes (BMMs), showed the number of osteoclasts was increased but the volume was smaller. They concluded that the knockdown of *Tcirg1* inhibited large-osteoclasts formation, by reducing calcium oscillation which inhibited the translocation of nuclear factor of activated T-cells 1(NFATc1).

### 3.3. ClC-7

The Chloride Voltage-gated Channel 7(*CLCN7*) gene encodes a multi-pass membrane protein that is located in the lysosome, late endosome, and the ruffled membrane of osteoclasts. Traditionally, in osteoclasts, ClC-7 acts as a 2Cl^−^/H^+^ antiporter and cooperates with V-ATPase to ensure the acidification of extracellular resorption lacuna, regulating the calcification and degradation of bone. Our previous investigation indicated that the knockdown of *clcn7* disrupts the balance of the TGF-β/BMP signaling pathway, causing the aforementioned craniofacial bone and tooth defects [3]. Peng et al. [75] showed that a novel *CLCN7* mutation contributed to the increased bone mass by increasing CD31^hi^EMCN^hi^ vessel formation and bone formation. These results reveal some novel insights into the pathogenesis and treatment of osteopetrosis with *CLCN7* mutations.

### 3.4. OSTM1

Osteopetrosis-Associated Transmembrane Protein 1(OSTM1), a type I trans-membrane protein, is luminal, highly glycosylated, and highly expressed in osteoclasts, and is located in the late endosome, lysosome, endoplasmic reticulum, and trans-Golgi network. An *OSTM1* mutation causes the most severe classification of osteopetrosis, OPTB5. Accumulating evidence that OSTM1 functions as an ancillary β subunit of the CLCN7 protein, supporting lysosome resorption and bone function [76]. A study proposed that *OSTM1* mutation leads to severe osteopetrosis by disrupting Wnt/β-catenin signaling [77]. Pata et al. [50] generated the first *Ostm1* mouse model with the human mutation, which showed faster and oversized osteoclasts. They suggested that *Ostm1* acts with a dual effect, regulating preosteoclast fusion and lysosomal trafficking and exocytosis of mature osteoclasts, which are essential for bone resorption. A recent study suggested that a loss of function in *Ostm1* impaired the localization and dispersion of secretory lysosomes. A deficiency of *Ostm1* caused oversized osteoclasts with few or poorly developed ruffled borders. Further, ex vivo studies revealed the absence of acidification of extracellular resorption lacunae, and that the release of TRAP and CTSK enzymes was impaired [46]. 

### 3.5. SNX10

Sorting Nexin 10 (SNX10) belongs to the sorting nexin family, targets endosomal membranes, and participates in endosomal segregation and trafficking by the PX domain. SNX10 is induced by RANKL and is essential for RANKL-induced osteoclast formation and function. SNX10 is localized in the endoplasmic reticulum and nucleus of osteoclasts, and usually binds to phosphatidylinositol phosphate, affecting endosomal function [78,79]. SNX10 affects the expression of the MMP9 protein in the late stage of osteoclast formation, but without affecting the activity of CTSK and TRAP in the early stage [78]. Furthermore, SNX10 positively regulates p38, JNK, and ERK phosphorylation in the signaling pathway of osteoclasts [80]. Recently, Stein et al. [63] established a R51Q SNX10 knock-in mouse model which exhibited fewer osteoclasts and the absence of ruffled borders on the osteoclasts. Further research has suggested that there is a SNX10-dependent fusion mechanism regulating the size and functionality of osteoclasts [81].

### 3.6. PLEKHM1

Pleckstrin Homology Domain-Containing Protein Family Member 1 (PLEKHM1), a modular cytoplasmic protein, is composed of two pleckstrin homology (PH) domains, a rubicon homology (RH) domain and a RUN domain. Accordingly, the loss of specific PLEKHM1 domains affects vesicle secretion, transport, and ruffled-border formation, reducing the bone resorption of osteoclasts, which leads to bone sclerosis [1]. Most research supports the idea that mammalian PLEKHM1 binds to Rab7, the HOPS complex, and GABARAP (gamma-aminobutyric acid receptor-associated) family proteins, which promotes the fusion of lysosomes with late endosomes and autophagosomes [1]. Surprisingly, Maruzs et al. [82] reported that the novel mutant alleles of Drosophila, *plekhm1* and *def8*, do not have obvious influences on autophagy in Drosophila.

### 3.7. LRP5

Low-Density Lipoprotein Receptor-related Protein 5 (LRP5) was successfully isolated from human osteoblasts, and mouse NIH 3T3 cells transfected with LRP5 showed increased cell proliferation [83]. LRP5 protein can affect bone accumulation through the Wnt pathway, controlling the expression of genes involved in osteogenesis [84]. Traditionally, osteopetrosis has been thought to derive from enhanced osteoblast activity due to LRP5 affinity. However, recently some studies have revealed that LRP5 deficiency leads to increased osteoclast activity and bone loss, which is quite different from the manifestation of osteopetrosis. Khrystoforova et al. [85] found that the loss of *lrp5* in zebrafish led to craniofacial deformities and low bone mineral density in adults. Mutants have increased TRAP staining, larger resorption areas, and developmental skeletal dysmorphologies, suggesting a higher resorptive activity of osteoclasts in the deletion of LRP5 signaling [85]. Sun et al. [86] reached a similar conclusion. 

### 3.8. CA II

Carbonic Anhydrase II (CA2) is highly expressed in mature osteoclasts, and the mutations in *CA2* cause a very mild osteopetrosis, OPTB3, with renal tubular acidosis and cerebral calcifications. CA2 catalyzes the formation of HCO3^−^ and H^+^. The generated H^+^ are extruded by V-ATPase and HCO3^−^ is taken up by the CLCN7/OSTM1 2Cl^−^/H^+^ antiporter. A CA2 abnormality leads to the production of fewer or abnormal osteoclasts. In the absence of osteoclasts, old bone is not destroyed when new bone is formed, resulting in thick and frail bones [87].

### 3.9. SLC4A2

Solute Carrier Family 4 Member 2 (SLC4A2) is highly expressed in osteoclasts, and is only expressed on lacunar membranes during its differentiation, and is up regulated with the maturation of osteoclasts [67]. SLC4A2 induces a large amount of Cl^−^ to flow into cells by participating in the exchange of extracellular HCO3^−^ and intracellular Cl^−^ in osteoclasts. On the one hand, the accumulation of a large amount of Cl^−^ reduces the intracellular pH and activates pH-sensitive cysteine protease, which mediates the dynamic tissue of podosomes in osteoclasts to promote the formation of actin bands and cell diffusion [88]. On the other hand, part of Cl^−^ is secreted into the lacunae of osteoclasts to maintain a low pH level for bone demineralization [67].

### 3.10. IKBKG

Genetic mutations of the inhibitor of Nuclear Factor Kappa-B Kinase, Regulatory Subunit Gamma (IKBKG), which is also known as the NF-κB essential modulator (NEMO), lead to OLEDAID. IKBKG belongs to the NF-κB family, and the NEMO protein encoded by IKBGB is a regulatory subunit of the IKK complex that is essential for osteoclast formation and interacts with c-Src in osteoclast progenitors. The IKK complex contains primarily IKK1, IKK2, and IKKγ (NEMO). The dominant negative form of c-Src (Src251) regulates the degradation of NEMO, thus inhibiting the signaling of NF-κB [89]. Nemo-binding domain peptides can weaken IKK complex assembly, namely the inhibition of IKK2 and IKK1 binding with IKKγ/NEMO, thereby inhibiting NF-κB activation and preventing RANKL-induced osteoclast generation [90].

## 4. Clinical Craniofacial Bone and Tooth Phenotypes of Osteopetrosis Patients

### 4.1. Craniofacial Characteristics of Osteopetrosis Patients

Osteopetrosis patients have typical craniofacial phenotypes, including changes in the shape and proportion of the facial skeleton, calvarial sclerosis, and jawbone abnormality. Changes in the shape and proportion of the facial skeleton is neatly illustrated by macrocephaly, microcephaly, ‘Space-alien face’ appearance, increased head circumference, frontal bossing, swelling of the cheek, facial asymmetry, and small mandibles. Calvarial sclerosis is a localized osteosclerosis, occurring in the skull base and skull roof. Changes in the shape and proportion of the facial skeleton and calvarial sclerosis happens in all 13 types of osteopetrosis. Jawbone abnormality includes osteomyelitis of the maxillary and mandible, mandibular prognathism, diffuse swelling and hypertrophy, and increased mandibular bone density, which mainly occurs in OPTA1, OPTA2, OPTB1, OPTB2, OPTB3, OPTB5, and OPTB7. Thus, sclerosis of the head only occurs in the skull, such as the base and the roof of the skull, while the jawbone has the characteristics of bone dysplasia due to its unique organizational structure. Abnormalities in the middle of the face include hearing impairment, vision abnormalities, and nasal dysfunction. Hearing impairment is mainly manifested in conductive hearing loss, auditory stenosis, and deafness, mainly occurring in OPTA1, OPTA2, OPTB1, OPTB3, and OPTB8. Visual abnormalities mainly include optic nerve atrophy, nystagmus, and progressive visual loss, which occur in OPTA1, OPTA2, OPTB1, OPTB2, OPTB3, OPTB4, OPTB5, OPTB7, OPTB8, and OPTB9. Nasal dysfunction is mainly manifested in poor nasal ventilation caused by the destruction of the nasal cavity bone, cicatricial squamous lesions at the base of the nasal bridge, nasal alar incitement, paranasal dyspnea, chronic rhinitis, and nasal congestion caused by nasal passage stenosis, mainly appearing in OPTA2, OPTB1, OPTB2, OPTB5, and OPTB7 (shown in Figure 1).

### 4.2. Dental Characteristics of Osteopetrosis Patients

Osteopetrosis patients have typical phenotypes such as alveolar bone abnormalities, dental abnormalities, and periodontal tissue abnormalities. Alveolar bone abnormalities manifest as abnormal alveolar bone permeability, defect, and necrosis, which mainly occur in OPTB1, OPTB2, and OPTB7. Dental phenotypes are characterized by malocclusion, dental crowding, malformed roots and crowns, caries, deciduous teeth retention, tooth development delay, and missing teeth. Some of these abnormalities also include abnormalities in tooth enamel and the pulp chamber. All the rest have dental phenotypes, except OPTA1, OPTB4, and OPTB8. Periodontal tissue abnormalities are mainly manifested as gingival swelling and gingival hypertrophy, mainly in OPTA3, OPTB5, and OPTB6 (shown in Figure 1).

## 5. Molecular Mechanism of Craniofacial Bone and Tooth Phenotypes in Osteopetrosis

The maintenance of bone homeostasis relies on a strict balance between bone formation by the osteoblast and bone resorption by the osteoclast. The pathogenic genes of osteopetrosis can regulate osteoclasts or the balance between osteoclasts and osteoblasts. Therefore, most scholars believe that the craniofacial bone, dental development, and the skeletal structure share similar biochemical and physiological properties, focusing on osteoblasts and osteoclasts. In other words, the mechanism for regulating the development of the craniofacial bone and teeth is consistent with the mechanism of bone development.

So far, there has been limited research on the regulatory mechanism of craniofacial bone and tooth development in osteopetrosis. In our previous study, we investigated the causation of craniofacial bone and tooth dysplasia in osteopetrosis with *CLCN7* mutations. *Clcn7* is expressed in ameloblasts, odontoblasts, and dental follicle cells (DFCs), and affects the differentiation of these cells [15]. We concluded that ClC-7 affects tooth development by targeting these cells, and regulates tooth eruption through the interaction between DFCs and osteoclasts by using the RANKL/OPG pathway [15]. Loss of *clcn7* function leads to zebrafish craniofacial cartilage defects and tooth defects, as well as lysosomal storage and the down-regulation of CTSK. ClC-7/CTSK further alters the balance of TGF-β-like Smad2 signals and BMP-like Smad1/5/8 signals, which might explain the typical craniofacial and tooth phenotypes in osteopetrosis [3]. It is widely known that OSTM1 is an auxiliary subunit of ClC-7 which maintains the activity and stability of ClC-7; on the other hand, it can regulate osteoclast fusion through the calcium–NFATc1 pathway [50], so OSTM1 may affect craniofacial bone and tooth development through these two paths. Huang et al. [38] reported that root formation and tooth eruption were defective in *RANKL^−/−^* mice. Molecular studies have showed that the activity of IGF/AKT/mTOR was decreased, which affected odontoblast differentiation and dental abnormalities in RANKL-deficient mice. RANK is expressed in the internal and external enamel epithelium as well as in dental mesenchyma; on the other hand, it is a specific receptor of RANKL [91]. So we speculate that RANK may affect craniofacial bone and tooth development by directly targeting these cells, or through a similar pathway as RANKL. Alkhayal et al. [92] performed a proteomic analysis of stromal cells from dental pulp from osteopetrosis with CA2 mutations, which revealed changes in multiple pathways, including MAPK, ERK1/2, PI3K, and integrin. It can be inferred that CA2 may affect tooth development by these pathways. There is no report about the relationship between craniomaxillofacial phenotypes and SLC4A2, but as an HCO3^−^/Cl^−^ anion exchanger, SLC4A2 may interact with CA2 [93], which needs further exploration. Khrystoforova et al. [85] generated *lrp5* knockout zebrafish, which exhibited craniofacial deformity, decreased bone mineral density, and skull dysplasia. The authors performed a transcriptome analysis of the cranium, which revealed the up-regulation of TGF-β, p38, MAPK, and mevalonate pathways, which regulate osteogenic signal transduction and osteoclast differentiation. Thus, it can be seen that LRP5 may affect craniofacial bone development by these pathways. On the other hand, Wnt/β-catenin signaling is required for early tooth morphogenesis, and LRP5 mainly regulates the WNT/β-catenin signaling pathway in osteoblasts [94,95], so LRP5 also may affect tooth morphology though this signaling pathway. The above research provides significant insights into the regulatory mechanisms of craniofacial bone and tooth development (shown in Figure 2).

## 6. Conclusions

Herein, we summarize the clinical characteristics, especially the craniofacial phenotypes of osteopetrosis and the underlying pathogenic genes (shown in Table 1). These osteopetrosis-causative genes play different roles in bone development, which directly lead to various gene-specific phenotypes. 

The clinical phenotypes of OPTA1 and OPTA2 are mild and patients are generally asymptomatic for a long time. OPTA1 is the only type of osteopetrosis that is not associated with fracture occurrence. Osteopetrosis can occur in addition to immunodeficiency and rickets in OPTA1. Seventy percent of cases of autosomal dominant osteopetrosis are OPTA2, which is the most common form of osteopetrosis. OPTB3 is characterized by osteopetrosis, renal tubular acidosis, and cerebral calcifications. OPTB1 is infantile malignant osteopetrosis, and the survival age of patients is generally less than ten years. OPTB4 is infantile malignant osteopetrosis and all patients die by the age of two. OPTB5 has the most severe clinical phenotype, and some patients die in the first few years, even within a year. OLEDAID is a rare form of the disease, with anhidrotic ectodermal dysplasia-associated immune deficiency, osteopetrosis and lymphedema.

Eleven genes have been identified as being involved in the pathogenesis of osteopetrosis and determining the genetic heterogeneity of osteopetrosis. Its pathogenesis can be divided into two types, one is directly related to osteoclasts, and the others are closer to osteoclast and osteoblasts. *TNFRSF11A*, *TCIRG1*, *SNX10*, *CLCN7*, *CA2*, *OSTM1*, *PLEKHM1*, and *SLC4A2* play essential roles in osteoclasts, such as differentiation, formation, and various activities. *IKBKG*, *LRP5*, and *TNFSF11* are mainly due to the interaction of osteoblasts and osteoclasts, which cause an imbalance between bone formation and bone absorption, resulting in osteopetrosis. In a word, notable progress has been made in the elucidation of the genetic bases of osteopetrosis. However, at least 10% of cases lack a molecular classification. So in some cases, the underlying pathogenetic mechanism still needs to be better elucidated.

Craniofacial and tooth abnormality were first reported in osteopetrosis in 1965 [96]. Our preliminary research has shown that over 84% of osteopetrosis patients had typical craniofacial and tooth phenotypes, including macrocephaly, frontal bossing, and dental abnormalities [3]. In this article, we detailed the craniofacial phenotype of each type of osteopetrosis according to the OMIM classification. To sum up, the gene-specific craniofacial phenotypes could be one of the important diagnostic criteria, or a useful auxiliary diagnostic criterion aid for osteopetrosis. Osteopetrosis is a hereditary bone disease commonly encountered by dentists. It is recommended that dentists pay attention to the craniofacial condition of osteopetrosis patients to reduce missed diagnoses.

In conclusion, we have reviewed the clinical features, types, and related pathogenic genes of osteopetrosis, and emphasized the craniofacial and dental abnormalities, and presented a molecular explanation of the craniofacial and dental anomalies present in osteopetrotic patients. The limitations of our study include not paying special attention to other phenotypes, such as neurodegeneration [97], leukocytosis [98], and hematological abnormality [98], which may be associated with the structural functional correlations of different mutations. New advances in the treatment of osteopetrosis were also introduced in detail, such as toxic doses of zoledronate lead and the radiographic signs of OPT mentioned by Whyte et al. [99]. MSC-seeded biomimetic scaffolds may be a cell-based therapy for *RANKL* osteopetrosis [100]. Expanded-circulating HSPCs is a novel cell source for the treatment of *TCIRG1* osteopetrosis [101]. All of the above findings are useful in guiding treatment decisions and deserve further investigation.

**Table 1 ijms-24-10412-t001:** Summary of causative genes and associated characteristics, especially craniofacial bone and tooth phenotypes of osteopetrosis.

Name	Abbreviation	OMIM	Inheritance	Main Characteristics	Craniofacial and Dental Characteristics	Mutation	Reference
Osteopetrosis, Autosomal Dominant 1	OPTA1	#607634	AD	Diffuse, symmetrical osteosclerosis No increased fracture rate Normal or even increased trabecular bone strength No ‘Rugger-jersey spine’, variable sclerosis No endobones, diffuse pelvic sclerosis	Conductive hearing loss Visual impairment Calvarial sclerosis Thickened cranial vault Headache Mandible hypertrophy and density increase	*LRP5*	[4,102,103,104]
Osteopetrosis, Autosomal Dominant 2	OPTA2	#166600	AD	Diffuse, symmetrical osteosclerosis Multiple fractures ‘Rugger-Jersey’ spine Hip osteoarthritis Endobones Bone marrow failure	Facial nerve palsy Increased head circumference Vision loss, severe, beginning in childhood Pronounced skull base sclerosis, increased thickness of the cranial wall Mandibular osteomyelitis Craniofacial and dental deformities Hearing impairment Nasal congestion caused by nasal passage stenosis	*CLCN7*	[7,9,98,105,106]
Osteopetrosis, Autosomal Dominant 3	OPTA3	#618107	AD	Hepatosplenomegaly Recurrent fractures with minor trauma Radiodense spine, osteophytes of vertebral bodies, ‘Sandwich’ vertebrae, ‘Bone-within-bone’ appearance of vertebrae Generalized increase in density of long bones Some patients have generalized osteopenia Anemia, thrombocytopenia	Thickened and sclerotic calvarium, localized osteosclerosis of the skull Early tooth loss, toothache, red and swollen gums Osteolytic area in the frontoparietal bone	*PLEKHM1*	[20,21]
Osteopetrosis, Autosomal Recessive 1	OPTB1	#259700	AR	Failure to thrive Hepatosplenomegaly Osteomyelitis Uniformly dense skeleton pathology Fractures ‘Bone-within-bone’ appearance ‘Sandwich’ vertebrae Coxa vara Splayed metaphyses Pancytopenia, anemia Hydrocephalus, seizures (tetany), cranial nerve palsies Intelligence impairment	Macrocephaly, frontal bossing Thick and dense skull, narrowness of neural and vascular foramina Facial paralysis Deafness Blindness, extraocular muscle paralysis, nystagmus, optic atrophy Dental caries, dentition defect, tooth loss, tooth development delay, tooth deformity, alveolar bone abnormality Cicatricial squamous lesion at the base of the bridge of the nose, nasal incitement The periosteum of the mandible is swollen	*TCIRG1*	[23,24,25]
Osteopetrosis, Autosomal Recessive 2	OPTB2	#259710	AR	Hepatosplenomegaly Osteomyelitis, especially of the mandible Osteosclerosis Multiple fractures Genu valgum Anemia, thrombocytopenia, pancytopenia, extramedullary hematopoiesis Normal intelligence	Mandibular prognathism, facial deformity, forehead bulge, increased head circumference Facial paralysis Early blindness, optic atrophy Chronic rhinitis due to narrow nasal airway Dental caries, dental anomalies, deciduous teeth retention, alveolar bone defects	*TNFSF11*	[32,33,34,54,107]
Osteopetrosis, Autosomal Recessive 3	OPTB3	#259730	AR	Failure to thrive Hepatosplenomegaly Osteosclerosis Recurrent fractures Muscle weakness, hypotonia Anemia Symmetrical cerebral calcifications, optic nerve pallor Intellectual impairment Mixed proximal and distal renal tubular acidosis, hyperchloremic hypokalemic metabolic acidosis	Facial features abnormalities, small mandibles Hearing loss Caries, malocclusion, retained deciduous teeth Visual impairment	*CA2*	[40,41,42,108,109,110,111,112,113,114]
Osteopetrosis, Autosomal Recessive 4	OPTB4	#611490	AR	Growth retardation Severe osteopetrosis Generalized increased bone density Fractures with minor trauma ‘Bone-within-bone’ sign Anemia, reticulocytosis, thrombocytopenia Increased trabecular size Thickening of matrix Abnormal medullary bone formation Narrowing of medullary space ‘Rugger-jersey’ spine Increased bone density in epiphyseal growth plates	Sclerosis, particularly of the base of the skull Visual impairment, pale optic discs, optic nerve atrophy	*CLCN7*	[6,45]
Osteopetrosis, Autosomal Recessive 5	OPTB5	#259720	AR	Growth failure Hepatosplenomegaly Generalized increase in bone density Thrombocytopenia, anemia, pancytopenia, coagulopathy Loss of corticomedullary differentiation Abnormal thickening of trabeculae No signs of active bone remodeling/resorption Sharp transition from compact to trabecular bone Persistent cartilaginous matrix Osseous deposition of woven bone Severe bone marrow failure Overwhelming infection Intense sclerosis of vertebrae Metaphyseal flaring Central and peripheral nervous system abnormal	Microcephaly, ‘harlequin mask’ or ‘space-alien face’ appearance, ‘prominent eyebrow’ appearance Intense sclerosis of cranium, hypoplastic mandible Visual impairment, roving nystagmus, exophthalmia, retinal depigmentation, pale optic discs, optic atrophy, narrowing of optic foramen Abnormal dental development, gum hypertrophy Forehead bulge Paranasal asthenia	*OSTM1*	[46,47,48,49,51,115]
Osteopetrosis, Autosomal Recessive 6	OPTB6	#611497	AR	Cortical sclerosis of pelvic bones Band-like sclerosis of vertebral endplates, ‘Rugger-jersey’ spine Nonhomogeneous sclerosis of metadiaphyses, “Erlenmeyer flask” deformity of distal femora and proximal tibiae	The skull thickens and hardens Tooth loss, toothache, red gums	*PLEKHM1*	[53]
Osteopetrosis, Autosomal Recessive 7	OPTB7	#612301	AR	Growth failure Hepatosplenomegaly Increased bone density Fractures Increase of bony and cartilaginous trabeculae Significant reduction of medullary space Thickened bone of vertebrae, ‘Sandwich’ vertebra Severe anemia Central and peripheral nervous system abnormal	Increased head circumference Destruction of maxillary alveolar bone, increasing cranial plate, persistent swelling of the cheek Poor nasal ventilation, bone destruction of nasal cavity Progressive visual loss, nystagmus, optic nerve atrophy Caries, several teeth did not erupt, root malformations, gingivitis	*TNFRSF11A*	[54,55,56,116,117]
Osteopetrosis, Autosomal Recessive 8	OPTB8	#615085	AR	Failure to thrive Hepatosplenomegaly Dense bones, narrowed medullary space due to encroachment of cortical bone Anemia, thrombocytopenia	Macrocephaly, open fontanel, frontal bossing Facial nerve palsy Vision loss, optic nerve atrophy Narrow auditory canal Sclerosis of semicircular canals Fully ossified ethmoid air cells, and sphenoid sinuses	*SNX10*	[61,62,118]
Osteopetrosis, Autosomal Recessive 9	OPTB9	#620366	AR	Increased overall bone density Increased bone fragility Progressive renal failure and hyperparathyroidism Cortical sclerosis in proximal femur Anemia Pulmonary stenosis	Significant sclerosis of cranial bones and skull base Visual impairment	*SLC4A2*	[66,67]
Anhidrotic ectodermal dysplasia associated with immune deficiency, osteopetrosis, and lymphedema	OLEDAID	#300301	XLR	Diffuse osteosclerosis, particularly of the cranial vault Iliac wings with “bone-within-bone” appearance Metaphyseal bands were observed for long-bone extremities Inability to sweat adequately Lymphedema	Missing or conical teeth Thin, sparse/absent hair	*IKBKG*	[2,68,69]

AD: autosomal dominant; AR: Autosomal recessive; XLD: X-linked dominant.

## Figures and Tables

**Figure 1 ijms-24-10412-f001:**
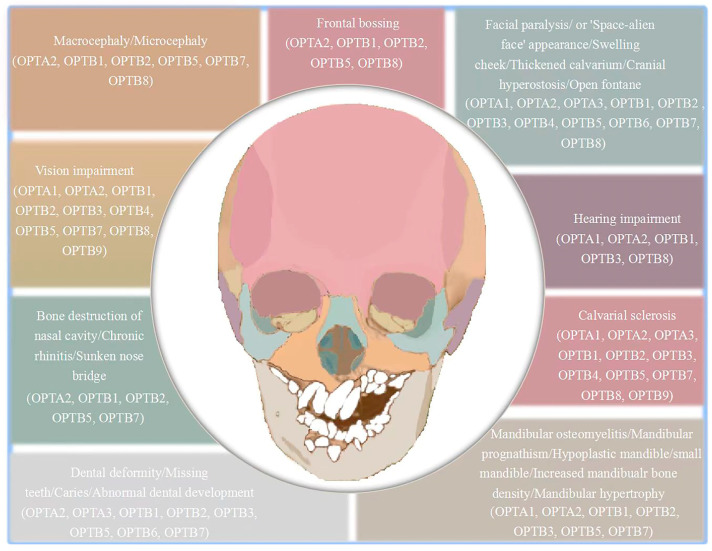
The craniofacial bone and tooth phenotypes with osteopetrosis.

**Figure 2 ijms-24-10412-f002:**
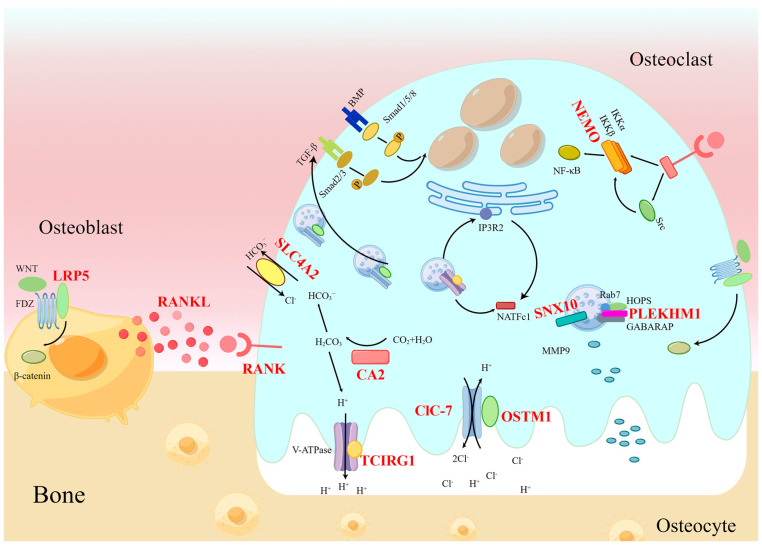
Simplified representation of molecules involved in osteoblasts and osteoclasts, showing the pathogenesis of craniofacial bone and tooth phenotypes in osteopetrosis. The well-known disease genes are in red (By Figdraw).

## Data Availability

Any data or material that support the findings of this study can be made available by the corresponding author upon request.

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
