# Peer review of "Molecular Mechanisms of Craniofacial and Dental Abnormalities in Osteopetrosis"

_ijms, 2023, doi:10.3390/ijms241210412_

Round 1
Reviewer 1 Report
This manuscript describes the main clinical characteristics as well as molecular pathways which are involved in the phenotype of osteopetrosis with emphasis on craniofacial and dental abnormalities. The authors provide a comprehensive description of the different types of the disease along with the causal mutations and the clinical symptoms referring to some useful case reports. In addition, they give a molecular explanation of the craniofacial and dental anomalies present in osteopetrotic patients.
The manuscript is well written and provides all the essential information for the topic. Some more details for the mechanisms that cause other than skeletal symptoms, e.g. visual impairment etc. would be useful. Otherwise, the manuscript can be accepted for publication.
Author Response
请参阅附件

Reviewer 2 Report
The review is interesting and well organised. This reviewer particularly appreciate the final table. However, I have some concerns that can improve the paper:
There are some particular papers that should be cited and discussed (JBMR 2021; 36:531-545. I suggest to read and enclosed the papers of Sobacchi C and/or Villa A. Furtermore, different recent paper also shoud be included: Wang X et al., etc.
A mention should also be spent for Whyte 2023 - DRUG-induced osteopetrosis
In the main page of the paper the authors' name with affiliations and detail lack.
Reviewer 3 Report
Osteopetrosis is a group of genetic bone disorders characterized by increased bone density and defective bone resorption. Osteopetrosis presents a series of clinical manifestations including craniofacial deformities and dental problems. The authors conclude that the telltale craniofacial and dental abnormalities are important for the dentists and other clinicians in the diagnose of osteopetrosis and other bone genetic diseases.
The authors carry out an exhaustive non-systematic review analyzing genotypic and phenotypic aspects related to osteopetrosis.
The figures introduced help to understand molecular aspects. It would be convenient for the authors to indicate: - The methodology for conducting the review - The limitations and strengths of paper
Reviewer 4 Report
This systematic review presents the major phenotypic molecular mechanisms in a quite heterogeneous number of different genetic disorders characterized by increased bone density and frailty focusing, in particular on craniofacial and dental defects.
The review is well-written and of interest.
A missing important point is the description of a relatively new described type of osteopetrosis: OPTB9 OMIM:# 620366 OSTEOPETROSIS, AUTOSOMAL RECESSIVE 9 which has recently been described (PubMed: 34668226) as being associated with SLC4A2 deficiency. Although the clinical description of these patients shows craniofacial defects, it is not obvious if dentition is involved in humans. One should remark, however, that slc4a2 inactivation in the mouse results in severe dental problems.
I think that in preparing a revised version of the paper it might be interesting to add some more information on the different existing mouse models in which OPTB genes have been either inactivated or mutated.
In essence, this review deserves to be published after completing some critical points.
The text is well-written and easy to read.
Round 2
Reviewer 2 Report
The authors addresses all my issues, thus the paper is suitable for publication.
Reviewer 4 Report
The authors have improved their manuscript which deserves now to be published.